# Greening Schoolyards to Improve Child Health: A Quasi-Experimental Study Protocol in Belgian and Dutch Primary Schools

**DOI:** 10.3390/ijerph22121805

**Published:** 2025-11-29

**Authors:** Bo H. W. van Engelen, Lore Verheyen, Bjorn Winkens, Michelle Plusquin, Robert Malina, Onno C. P. van Schayck

**Affiliations:** 1Department of Family Medicine, Care and Public Health Research Institute (CAPHRI), Maastricht University, P.O. Box 616, 6200 MD Maastricht, The Netherlands; onno.vanschayck@maastrichtuniversity.nl; 2Centre for Environmental Sciences, Hasselt University, 3590 Diepenbeek, Belgium; 3Department of Methodology and Statistics, Care and Public Health Research Institute (CAPHRI), Maastricht University, P.O. Box 616, 6200 MD Maastricht, The Netherlands; bjorn.winkens@maastrichtuniversity.nl; 4Environmental Economics Research Group, Centre for Environmental Sciences, Hasselt University, 3590 Diepenbeek, Belgium

**Keywords:** green schoolyards, physical activity, child health, cognition, well-being, BMI z-score, study protocol

## Abstract

Background: Childhood obesity and mental health problems are major public health concerns worldwide. Early-life exposure to green spaces has been shown to promote physical activity, reduce obesity risk, and improve cognitive and emotional development. Schoolyards offer a unique opportunity to promote health, as children spend a large proportion of their time at school. Methods: This quasi-experimental protocol study investigates the effects of transforming gray schoolyards into biodiverse green spaces on children’s health and well-being. Four primary schools in Limburg (Belgium and The Netherlands) were recruited: two intervention schools and two control schools. Children aged 7–12 years were enrolled, with baseline data collected in November 2021 and follow-up measurements scheduled every six months until November 2023. Outcomes include body mass index (BMI) z-score (primary outcome), waist circumference, diet, cognitive performance, psychological well-being, biodiversity knowledge, and physical activity. Data will be analyzed using linear mixed models, and cost-effectiveness analyses will be performed. Expected Results: Improvements in BMI z-scores, cognitive functioning, dietary behavior, and psychological well-being are expected among children in green schoolyards compared to those in control schools. Increased biodiversity awareness and reduced exposure to black carbon are also anticipated. Conclusions: This study is designed to provide evidence on the health impacts of greener schoolyards and contribute to strategies for promoting child development through environmental interventions.

## 1. Introduction

Childhood obesity has emerged as a critical global public health challenge [1]. The development of overweight and obesity in youth is closely linked to lifestyle behaviors, particularly inadequate physical activity (PA) and unhealthy dietary patterns [2]. Recent estimates from the World Health Organization indicate that in 2022 more than 390 million individuals aged 5–19 years were classified as overweight, including 160 million with obesity. This number reflects a 12% increase in prevalence among children and adolescents since 1990, signaling a rapid upward trend [3]. Such figures are cause for concern, as excess adiposity significantly elevates the risk of cardiometabolic conditions, including cardiovascular disease, type 2 diabetes, and several cancers [2,4]. Beyond physical health, obesity coincides with a broader rise in mental health difficulties among young people worldwide [5], which can adversely affect educational outcomes, overall well-being, and long-term development. According to UNICEF, approximately 86 million adolescents aged 15–19 years and 80 million children aged 10–14 years currently experience mental disorders [6].

Exposure to natural environments during childhood has been associated with multiple dimensions of healthy development. Empirical studies indicate that contact with green space can support emotional well-being, enhance neurocognitive functioning, and foster positive social-behavioral outcomes [7,8,9,10]. In addition, regular interaction with green environments has been linked to favorable physical health indicators, including higher levels of physical activity [8,10,11], improved respiratory function [12], lower body mass index [10,13], and reduced risk of overweight and obesity [8,10,13]. Because lifestyle patterns established early in life often continue into adulthood, interventions that encourage healthy habits during childhood are considered particularly impactful [14]. Promoting such behaviors from a young age has the potential to contribute to both improved health trajectories and academic performance later in life [15]. Given that children spend a substantial portion of their daily lives at school, creating greener schoolyard environments offers a promising strategy to strengthen their engagement with nature and support healthier developmental outcomes. Schools are considered important environments for promoting healthy habits at an early age for several reasons. First, schools can reach all children from diverse backgrounds in a relatively short period. Second, the structured school environment facilitates the implementation of interventions. Moreover, children spend a significant portion of their day here, often with one or two meals. Schools also provide opportunities for PA and health education [16,17]. By constantly exposing children to elements of a healthy lifestyle at school, unhealthy behaviors can be changed [6,17].

The Healthy Primary School of the Future (HPSF) is a Dutch initiative integrating structured physical activity and school-provided healthy meals into the daily routine. Previous evaluations showed favorable effects on children’s lifestyle behaviors and BMI z-scores. More details on the HPSF framework and outcomes have been published elsewhere [15,18].

Building on the foundation of the Healthy Primary School of the Future and existing evidence on the value of natural environments, the Green Healthy Elementary School of the Future initiative was conceived to expand the focus of school health promotion. This program aims to evaluate how transforming conventional playgrounds into biodiverse, nature-rich spaces influences children’s physical and mental health, particularly through opportunities for sensory and exploratory play. The intervention is grounded in the socio-ecological model of school health promotion [19], which emphasizes that behaviors are shaped not only by individual characteristics, but also by broader contextual factors, including social relationships, cultural norms, economic circumstances, and physical environments. According to Bell and Dyment, a green schoolyard is defined as an extracurricular environment where natural elements such as trees, flowers, sand, water, grass, hills, and shrubs are combined to create a more attractive schoolyard and improve the quality of children’s (play) experiences [20].

Several theoretical frameworks help explain why exposure to biodiverse natural environments may benefit children’s health and development. The Attention Restoration Theory (ART) proposes that natural settings support the recovery of directed attention by providing softly fascinating stimuli, which may improve cognitive functioning and learning outcomes [21]. Similarly, the Stress Reduction Theory (SRT) posits that contact with natural environments can evoke psychophysiological stress recovery, thereby enhancing emotional well-being and reducing mental fatigue [22,23]. In addition, the Biodiversity Hypothesis suggests that exposure to diverse natural microbiota may beneficially influence immune function through microbial-mediated training pathways, with potential implications for respiratory health and environmental exposure processing [24,25,26,27,28,29,30,31,32]. These frameworks collectively support the inclusion of outcomes such as cognitive performance, psychological well-being, PA, and biomarkers of environmental exposure, including black carbon in the present study.

Despite the growing evidence for the benefits of green spaces on children’s development, substantial knowledge gaps remain. Most existing studies are cross-sectional or short-term and focus primarily on general neighborhood greenness rather than on biodiverse, intentionally designed school environments [7,8,9,10,11,19,33,34,35,36]. Few studies have simultaneously assessed ecological features, such as quantified biodiversity, together with cognitive outcomes, psychosocial well-being, PA, and environmental exposure biomarkers. Importantly, no previous schoolyard greening protocols have integrated biodiversity assessments with objective measures of black carbon as an internal exposure marker or combined these with detailed cognitive testing in a longitudinal framework. As a result, the mechanisms through which biodiverse schoolyards may influence child health, including potential pathways described in the Attention Restoration Theory, Stress Reduction Theory, and Biodiversity Hypothesis, remain insufficiently understood [19,21,22,23,33,34,35]. The present study addresses this gap by employing a multidisciplinary, longitudinal protocol that captures ecological, behavioral, physiological, and environmental indicators within the same analytical framework.

In the current explorative study, we aim to investigate whether renovating a schoolyard to increase biodiversity has an effect on children’s BMI Z-score, waist circumference, dietary behaviors, health markers, cognitive performance, knowledge and appreciation of biodiversity, and (school) well-being, compared to children in control schools. Additionally, we will compare the effect of a green schoolyard on children’s PA with the partial HPSF.

## 2. Materials and Methods

### 2.1. Study Design and Setting

This study follows a quasi-experimental design involving two intervention schools undergoing schoolyard greening and two comparison schools without environmental changes. All participating schools are primary schools situated in the Province of Limburg of Belgium and The Netherlands (Figure 1). These provinces are characterized by a low SES, compared to the means of the countries [36,37]. ClinicalTrials.gov identifier (NCT number): NCT04898439. Baseline data collection was completed prior to the start of the intervention, with all baseline measurements conducted in November 2021.

School and subject recruitment. Two schools actively participate in the process of greening the schoolyard (one in Belgium and one in The Netherlands). Two other schools in the nearby region are control schools (one in Belgium and one in The Netherlands). Randomization was not possible due to the necessity for complete voluntary cooperation and participation of the intervention schools in greening the schoolyard. Because participation in the greening intervention required voluntary school commitment and alignment with construction timelines, randomization was not feasible. Therefore, potential confounders at both the child and school level will be measured and adjusted for in the linear mixed models. Baseline measurements were completed in November 2021 (T0). Follow-up measurements were planned each half-year for two years until November 2023 (T6: May 2022, T12: November 2022, T18: May 2023, and T24: November 2023). In The Netherlands, not all measurements could be taken due to practical reasons.

Intervention school recruitment. In early 2021, a newspaper call in The Netherlands recruited nine schools from Limburg to green and diversify their schoolyards. These schools were interviewed, and one was selected to participate as an intervention school on the basis of feasibility criteria and the comparability of the schoolyard size and characteristics. Selection of the Belgian intervention and control school was performed by the ‘Provinciaal Natuurcentrum Hasselt’ based on their call to green schoolyards and make schoolyards more biodiverse the comparableness of the schoolyards and number of enrolled students. The recruitment of the intervention schools was completed in September 2021.

Control school recruitment. The schools that voluntarily agreed to participate were selected on the basis of the following inclusion/exclusion criteria: a high pavement-to-greenery ratio and a location in the proximity of the intervention schools with comparable populations and SES backgrounds.

Subject recruitment. Children were recruited for the baseline measurement through informational brochures sent via mail and reminders from the school staff. Furthermore, the research team visited all classrooms to provide information about the study and encourage participation. All children aged 7–12 from both intervention and control schools were eligible for inclusion in this study. Before the baseline examination, written consent from the parents was obtained and children gave their oral permission to participate. The enrollment of children for the initial measurement was concluded in November 2021.

Intervention. Both intervention schools underwent a greening procedure increasing biodiversity, in which they adjusted the gray schoolyard into a green biodiverse schoolyard consisting of (i) the creation of adventurous and diverse play areas, (ii) the establishment of green rest areas, (iii) the planting of flowers, shrubs, hedges and/or trees, and/or (iv) variation in activity zones such as farm animals, pools or vegetable gardens.

Because the greening intervention was implemented by two different schools and included varying components, such as water features, gardens, and diverse vegetation, some heterogeneity in intervention intensity is expected. As each country (Belgium and The Netherlands) includes one intervention and one control school, the correction for country as fixed factor captures unmeasured differences between schoolyards.

### 2.2. Outcome Parameters

Table 1 shows an overview of all outcome measures collected in children and parents/caregivers. The administration of questionnaires that were completed by the children were taken collectively during regular classes. The children independently read and completed the questionnaires. Throughout this process, at least one researcher was present to oversee and address any queries. The anthropometric measurements were taken individually. Tailored questions, accompanied by various graphical illustrations, were utilized in the lunch and general questionnaire. The cognitive performance of children was assessed during regular classes in a separate room. To prevent excessive burden on children and school staff, the majority of child-related measurements were scheduled within a regular school week.

Personal characteristics questionnaire. The parents filled out a questionnaire before the baseline examination to collect information about parental education, parity, ethnicity, parental occupation, income, household smoking behavior, illness, socio-economic demographics, birth date of the child, sex of the child, and the use of medication. Completion of privacy-sensitive questions, such as parental BMI, socio-economic status, and disease status, were optional.

Strengths and Difficulties Questionnaire (SDQ). At each examination, a parent completed the Strengths and Difficulties Questionnaire (SDQ) to assess the child’s emotional and behavioral well-being [38]. The validated SDQ includes five subscales: emotional symptoms, conduct problems, hyperactivity/inattention, peer relationship problems, and prosocial behavior. The first four subscales contribute to a total difficulty score, with lower scores indicating fewer social-emotional difficulties. Responses are scored on a three-point scale (0–2).

(Psycho) social functioning. The self-reported, Dutch-validated KIDSCREEN-52 questionnaire assesses quality of life in children and adolescents across 10 domains, including physical and emotional well-being, self-perception, relationships, school life, autonomy, financial well-being, social support, bullying, and psychological well-being [39,40]. Each of the 52 items is weighted and summed into raw subscale scores, which are then converted into standardized T-scores, with higher scores indicating better quality of life. Additionally, attention bias was assessed using the Tobii po Nano eye tracker (Tobii, Stockholm, Sweden), based on emotion recognition from facial expressions as described by Goeleven et al. [41]. Bias was calculated as the ratio of first fixation duration on ‘happy’ versus ‘anxious’ and ‘sad’ emotional faces.

### 2.3. Food Intake

Food intake is assessed using a combination of a food frequency questionnaire and a dietary recall tool, which are completed by both children and parents. “The food intake questionnaire, tailored specifically for this target group, is based on the Short Dutch Questionnaire to Measure Total and Saturated Fat Intake and the Short Dutch Fruit and Vegetable Intake Questionnaire developed by Van Assema et al. [42,43].” It includes items on fruit and vegetable consumption, soft drinks, sports and energy drinks, and snacks. Dietary recall is employed to evaluate the composition of breakfast and lunch, with questions about lunch intake asked in the classroom immediately after the lunch break. To prevent influencing children’s dietary habits, they are not informed in advance that they will be questioned about their food consumption.

### 2.4. Knowledge and Affinity to Biodiversity

The custom-designed biodiversity questionnaire (Appendix A) generated a score that was used to determine knowledge and affinity for nature and biodiversity. A higher score means a higher affinity for nature.

### 2.5. Anthropometric Measurements

Weight is assessed with a precision of 0.1 kg using the Weighing Scale 803 (Seca, Hamburg, Germany), while height is measured with a Stadiometer 213 (Seca, Birmingham, UK) to the nearest 0.1 cm. Children undergo measurements wearing light clothing and without shoes. BMI Z-scores are computed using Dutch and Belgium reference values [44]. The primary outcome is the absolute change in BMI Z-score, aiming for BMI scores closer to national and international standards, resulting in reduced BMI for overweight and obese children and increased BMI for underweight children. Hip and waist circumferences were gauged with a measuring tape (model 201, Seca, Hamburg, Germany) to the nearest 0.1 cm, following the World Health Organization’s assessment protocol [45]. Each anthropometric measurement was conducted twice, with a third measurement implemented if the discrepancy between the initial two measurements surpassed predefined limits (weight ≥ 0.2 kg, height ≥ 0.5 cm, hip and waist circumference ≥ 1.0 cm). These outcomes were then averaged.

### 2.6. Physical Activity Behavior

PA levels were objectively evaluated using the Actigraph accelerometer (Actigraph, GT3X+, Actigraph, Pensacola, FL, USA). The monitor was affixed to the right hip using an elastic band, and all children were instructed to wear the device continuously for seven days. The device should be worn throughout the day, excluding sleeping hours and activities involving water (such as swimming, bathing, or showering). Concurrently, parents completed a brief activity diary in the same week, detailing their child’s PA, swimming behavior, and any exceptional circumstances (e.g., the child’s illness).

Non-wear time was defined as ≥60 consecutive minutes of zero counts, allowing for interruptions of up to 2 min of counts between 0 and 100. A valid day was defined as a minimum of 480 min (8 h) of wear time, and children were included in analyses if they had at least three valid weekdays and one valid weekend day. Days not meeting these criteria were classified as invalid and excluded from analyses. When accelerometer data were incomplete but accompanied by diary notes explaining non-wear reasons, these days were recorded but remained excluded from accelerometer-derived outcomes. Average daily counts, sedentary time, and moderate-to-vigorous PA (MVPA) were derived only from valid days.

The Physical Activity Questionnaire for older children (PAQ-C) was surveyed to assess overall levels of PA during greening [46]. The PAQ-C is appropriate for elementary school children (approximately 8–14 years old) who are currently in a school system and have regular recess as part of their school week. It is a validated questionnaire with a 7-day look-back period that assesses general moderate to vigorous physical activity levels measured during the school year. The PAQ-C provides a summary PAscore derived from nine items each rated on a 5-point scale with 1 being the lowest activity score and 5 being the highest activity score. This questionnaire was administered at T0, T12 and T18.

### 2.7. Cognitive Performance

To assess cognitive effects, five computerized tests were administered using the Mindsware test manager (Bureau Mindsware, Den Helder, The Netherlands), guided by trained staff [47]. The continuous performance test (CPT) evaluated attention and impulsivity by requiring responses only to specific stimulus combinations (e.g., X after A). The symbol digit modalities test (SDMT) assessed information processing via symbol-number matching. The SPANNE test measured short-term memory by having children repeat number sequences forward and backward. The signal detection test gauged visual processing speed by identifying deviant symbols in a series. Lastly, the STROOP task tested cognitive control by having participants name the ink color of incongruent color words.

### 2.8. Particulate Matter Determination in Urine

The exposure to fine particulate matter was measured using an internal exposure marker, black carbon, employing a specific and sensitive detection technique. Measurement was based on the white light generation of carbon particles under femtosecond pulsed illumination, as previously developed, validated, and applied in human biomonitoring studies [48,49]. This in-house developed and validated technique enables label-free measurement of black carbon concentrations in urine, reflecting the internal accumulation of medium to chronic exposure to traffic-related air pollution. Urine samples collected per grade level were pooled into one sample per assessment time point, both for the control and intervention schools. 100 μL urine sample was loaded per imaging chamber, constructed by placing a coverslip (24 × 24 mm) onto a microscope slide (75 × 25 mm), joined with 100 μm thick double-sided tape. The urine-filled imaging chambers were sealed airtight to prevent dehydration. Images of the urine samples were captured at room temperature at nine different locations within the imaging chamber using a Zeiss LSM880 confocal microscope (Carl Zeiss Microscopy GmbH, Jena, Germany). A peak-finding algorithm was applied to determine the number of black carbon particles in the images. The average amount of detected particles in the various images was normalized to the osmolality of the measured samples. Ultimately, the result was expressed as the number of detected black carbon particles per milliliter of urine. A formal limit of detection cannot yet be established for this optical method, as no certified reference materials with known black carbon particle counts exist. Nevertheless, prior studies demonstrate that the technique is sufficiently sensitive and reproducible to detect differences in internal black carbon exposure in human samples [48,49].

### 2.9. Number of Participants

Due to the explorative nature of the study (hypothesis generating), no detailed sample size calculations are performed. The aim is to include 150–200 participating children from the participating schools in The Netherlands and Belgium. To account for missing data (assumed to be around 20%), the effective sample size for the analysis might vary between 80 and 120 children (40 to 60 per group). Based on 80% power and a significance level α of 0.05, standardized effect sizes (Cohen’s) between 0.52 and 0.63 can be detected with these expected sample sizes. This study was approved by the commission for Medical Ethics of Hasselt University (CME2021/042) and is a part of the original HPSF trial (METC-Z no. 14-N-142).

### 2.10. Data Analysis

Linear mixed model analysis will be employed to evaluate the longitudinal impact of the intervention on the primary outcomes, absolute BMI Z-scores, change in cognitive function, change in well-being, and change in black carbon levels. Data will be analyzed using linear mixed models, performed in IBM SPSS Statistics for Windows (version 28.0.1.1, Armonk, NY, USA: IBM Corp). This approach addresses correlations within individuals inherent in repeated-measures research designs. Its advantage lies in handling missing values naturally through a likelihood-based method, assuming data are missing at random.

The fixed part of the model incorporates group (intervention vs. control), time (measurement time points), the interaction term group*time, and country (Belgium/The Netherlands). Baseline variables linked to missing data or outcomes will also be included for unbiased and precise results. To account for the correlation between repeated measurements several covariance structures (including random effects if necessary) will be considered. The best covariance structure for the repeated measurements will be chosen based on Akaike’s Information Criterion. The analysis will encompass all participants with at least one outcome measurement. Because several measurements were collected only in Belgian or only in Dutch schools, the resulting missingness is structured and not at random. Linear mixed models are used because they accommodate unbalanced data and allow all available observations to be included without listwise deletion. Country will be included as a fixed effect in all models to control for systematic differences between the Belgian and Dutch school contexts. For outcomes collected in only one country, analyses will be conducted within that specific subsample and reported separately. In addition, sensitivity analyses will be performed to examine whether excluding country-specific measures affects the direction or magnitude of findings for outcomes measured in both countries.

No school-level random effects were added as the number of schools is limited to four (two in Belgium, two in The Netherlands). Since each country only includes one intervention and one control school, including country as fixed factor accounts for the school effect. Distinct statistical methods will address the other research inquiries. Longitudinal effects on numerical quantitative outcome variables will be evaluated employing the identical linear mixed models utilized for the primary outcome. Categorical outcome variables, on the other hand, will be scrutinized through a logistic mixed-model analysis technique (or generalizing estimating equations (GEE)), adopting a model akin to the one delineated for the primary study parameter.

To account for the correlation between repeated measurements several covariance structures (including random effects if necessary) will be considered. The model-fit will be evaluated using the −2log likelihood, where the best covariance structure for the repeated measurements will be chosen based on Akaike’s Information Criterion.

## 3. Discussion

The Green Healthy Elementary School of the Future initiative addresses rising childhood obesity rates and the benefits of early exposure to green environments.

In this quasi-experimental study, the effects of transforming schoolyards into biodiverse green spaces on children’s physical and mental health, cognitive development, and well-being are examined.

There is strong evidence linking green spaces to better cognitive performance in children, including improved attention, academic outcomes, and stress reduction [50]. Exposure to nature is associated with lower risks of mental disorders in urban populations [51], supported by the Attention Restoration Theory (ART) and Stress Recovery Theory (SRT), which explain how nature reduces cognitive fatigue and stress [21,22,23].

An increase in schoolyard biodiversity is achieved through the intervention, which is associated with health benefits [22,23,24,25], supporting the biodiversity hypothesis: exposure to diverse species may enhance immune function through microbial diversity [26,27,28,29,30]. The biophilia hypothesis suggests that humans are naturally drawn to nature, benefiting their health [52,53,54], while the dilution effect hypothesis posits that biodiversity can reduce disease transmission [55,56,57].

By encouraging active, nature-based play, an increase in PA is promoted, and may reduce obesity risk. Previous studies link green spaces to increased PA and healthier BMI in children [58,59,60,61]. The socio-ecological design integrates natural elements to foster imaginative play and improve social behavior.

A key advancement of this study is its ability to explore potential mechanisms underlying these health effects by integrating ecological, behavioral, cognitive, and environmental exposure data within a single protocol. Whereas most previous school-greening studies focused solely on behavioral or psychological outcomes, this study combines detailed biodiversity assessments, objective PA measurements, cognitive testing, and biomarkers of internal black carbon exposure. This multidimensional approach allows examination of whether increased biodiversity is associated with changes in cognitive functioning via restorative and stress-reduction pathways, and whether greener schoolyards may mitigate children’s internal exposure to airborne pollutants by altering microenvironmental conditions. By linking ecological features of the schoolyard to both cognitive and physiological markers, this study provides the opportunity to investigate mechanistic pathways that have rarely been tested empirically in school settings.

Though limited by its non-randomized design, the inclusion of schools from Belgium and The Netherlands enhances generalizability. A comprehensive set of outcome measures—anthropometric data, cognitive assessments, questionnaires, and activity monitoring—supports a robust analysis. Due to the modest sample size, the study is considered an exploratory study aimed at informing future, larger-scale research.

Challenges include school-specific logistical barriers, reliance on some self-reported data, and the need for long-term maintenance. Future studies should incorporate more objective data and extended follow-up periods.

While previous research has evaluated green schoolyards [58,59,60,61], the present study is distinguished by its focus on biodiversity and novel outcomes such as black carbon exposure and cognitive functioning. Comparisons with a comprehensive school health program that includes PA and nutrition [18] offer deeper insights into the potential of biodiverse schoolyards.

Several limitations should be considered. First, the non-randomized design restricts the ability to infer causality and introduces potential confounding. Second, sample size limitations reduce statistical power and restrict subgroup analyses. Third, some measurements, particularly dietary behavior and well-being, rely partly on self-report, which may introduce reporting bias. Additionally, variations in data collection between countries (e.g., missing time points in The Netherlands) may lead to differences in data completeness.

Future research should aim to employ randomized or matched cohort designs, increase sample sizes, and include longer-term follow-up to evaluate sustained effects. Incorporating more objective measurements (e.g., continuous activity monitoring, environmental sensors, biological markers) will strengthen the evidence base. Future studies could also explore differential effects by socioeconomic status, baseline nature affinity, or schoolyard design features.

Study findings will be disseminated through peer-reviewed scientific publications, presentations at public health and environmental health conferences, and reports to participating schools and local authorities. Results will also be communicated to stakeholders involved in school policy, urban planning, and environmental education to support evidence-based decisions and promote the integration of biodiverse schoolyards into broader health-promotion strategies.

## 4. Conclusions

Ultimately, this study may guide future school-based health strategies by highlighting how greening schoolyards can support children’s development, reduce obesity, and foster healthier, more sustainable learning environments. By generating evidence on the combined effects of biodiversity, PA opportunities, and reduced environmental exposures, the findings have the potential to inform policy decisions regarding the design of school environments, integration of nature-based interventions into public health programming, and prioritization of green infrastructure in educational settings. If effective, the intervention model may serve as a scalable and cost-efficient approach that municipalities and school boards can implement to promote child health, enhance learning conditions, and contribute to broader environmental and sustainability goals.

## Figures and Tables

**Figure 1 ijerph-22-01805-f001:**
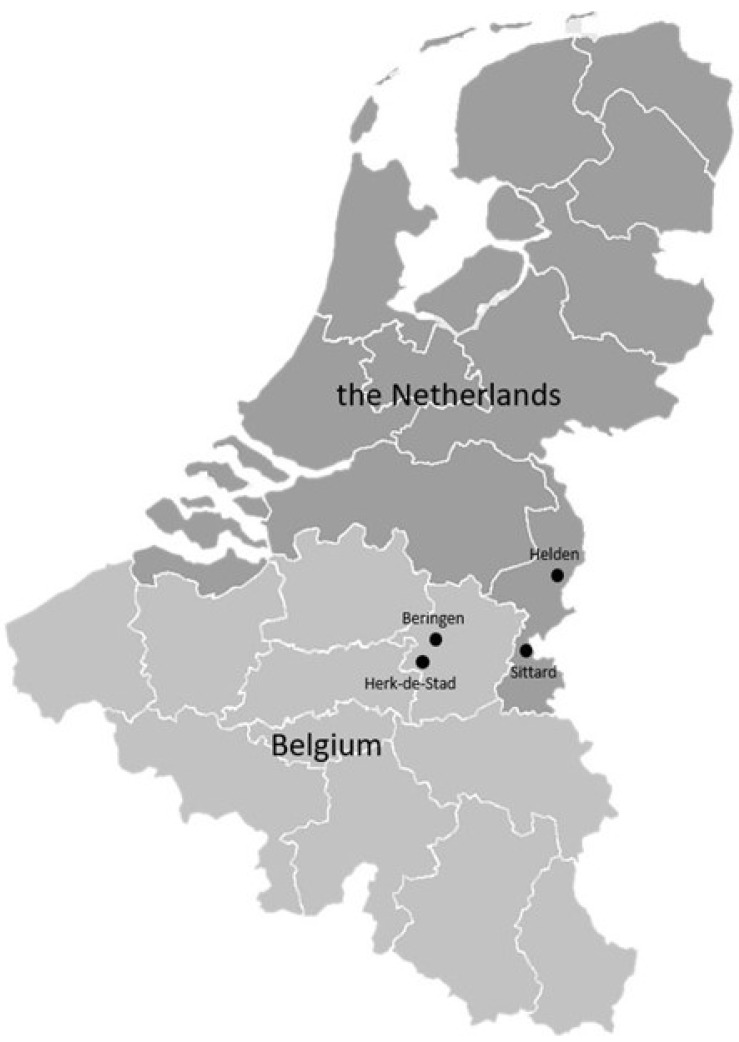
Map of Intervention and Control School Locations in the Study Region.

**Table 1 ijerph-22-01805-t001:** Overview of questionnaires and measurements for data collection measures within the Green Healthy Elementary School of the Future initiative.

Data Collection Activity	Parents	Children	Time Point
General questionnaire			
Personal data (birth data, sex, ethnicity, addresses, parity, SES, education level of the parents)	x		T0
Sleep ^1^	x		T0, T6, T12, T18
School absenteeism ^1^	x		T0, T6, T12, T18
Household smoking habits	x		T0, T6, T12, T18, T24
Visits to GP, disease status ^3^	x		T0, T6, T12, T18
Quality of life (parental PedsQL) ^1^	x		T0, T6, T12, T18
PA behavior	x		T0, T6, T12, T18
Food + water consumption, habits and preferences ^1^	x	x	T0, T6, T12, T18
Strengths and difficulties questionnaire (SDQ)	x		T0, T6, T12, T18, T24
Kidscreen-52		x	T0, T6, T12, T18, T24
Physical Activity Questionnaire for Children (PAQ-C)		x	T0, T12, T18
Bullying behavior		x	T0, T6, T12, T18, T24
Knowledge, perception and valuation of biodiversity		x	T0, T12, T18, T24
Height		x	T0, T12, T18, T24
Weight		x	T0, T12, T18, T24
Waist/hip circumference		x	T0, T12, T18, T24
Physical activity using Accelerometer (Actigraph, GT3X+, Actigraph, Pensacola, FL, USA) ^2^		x	T0, T12, T18
Cognitive tasks and eye-tracking		x	T0, T6, T12, T18, T24
Particulate matter determination in urine ^3^		x	T0, T6, T12, T18, T24

^1^ Measurements only performed in Dutch schools. ^2^ Measurements only performed in intervention schools. ^3^ Measurements only performed in Belgian schools. Due to practical reasons T6 and T24 measurements could not be performed in Dutch primary schools. T0: November 2021, T6: May 2022, T12: November 2022, T18: May 2023, and T24: November 2023.

## Data Availability

The datasets used and/or analysed during the current study are available from the corresponding author on reasonable request.

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
