# Peer review of "Greening Schoolyards to Improve Child Health: A Quasi-Experimental Study Protocol in Belgian and Dutch Primary Schools"

_ijerph, 2025, doi:10.3390/ijerph22121805_

Round 1

Reviewer 1 Report

Comments and Suggestions for Authors

If we start from the beginning, the first impression comes from the very title, that is, the definition of the problem. I can say with pleasure that the topic is excellent, that is, the problem being investigated is indeed important in the context of children's health. On the other hand, the authors of this study point to the fact that a lot of things are still in the hands of people who can really influence health. An example of this is this study of school greening protocols. In this sense, I believe that the study has good potential among readers, both experts and others.

Already in the abstract, in some part, it would be good to insert a sentence that it is a protocol study. It is true that this mark is only in the left corner above the title, but I think that potential readers can easily overlook it. In this sense, I recommend that it is also found in keywords. Also when it comes to the abstract in line 29 Results: We hypothesize that children in green schoolyards will show improve... Eliminate sentences in which it is spoken in the third person use impersonal or passive constructions.

It is also necessary to mention, both in the abstract and in the data analysis, which software program will be used and, more precisely, which statistical analysis.

The introductory part of the work is well conceived and the authors successfully introduce the reader to the issues of the work and support the stated claims with adequate references.

In the Study design and setting part, it should be noted that baseline research was done.

In the outcomes parameters section, the researchers correctly chose the potential instruments that they have or will use in their study.

In line 182 and 183 "The food intake questionnaire, tailored specifically for this target group, is based on two questionnaires developed by Van Assema et al. [27,28]." I should immediately put the names of those two questionnaires that are in the mentioned reference.

Anthropometric measurements, Physical activity behavior. Cognitive performance. Particulate matter determination in urine. In all these parts, the work tests are well chosen and measure what the authors envisioned.

In the Discussion part, it is necessary to clearly emphasize with the subsection Limitations and Future Research and after that another Dissemination of Results

Anthropometric measurements, Physical activity behavior. Cognitive performance. Particulate matter determination in urine. In all these parts, the work tests are well chosen and measure what the authors envisioned.

As can be seen from the suggestions, the proposal is to do a major revision

Author Response

Response to Reviewer 1

We thank the reviewer for the thoughtful and constructive comments. We have addressed each point in detail below and revised the manuscript accordingly.

Comment 1

The topic is excellent and relevant for children’s health; the study has strong potential.

Author Response:
We thank the reviewer for the positive evaluation of the study’s relevance and potential impact.

Comment 2

Add a sentence in the abstract noting that this is a protocol study. Readers may overlook the “Study Protocol” tag. Also include this in the keywords.

Author Response:
We agree. The abstract has been revised to explicitly state that the manuscript describes a study protocol (added in sentence 20). In addition, “study protocol” has been added to the list of keywords.

Comment 3

Eliminate third-person phrasing and use impersonal or passive constructions.

Author Response:
Revisions have been made throughout the abstract and discussion to ensure neutral, impersonal phrasing consistent with protocol-writing guidelines.

Changes in abstract:

  • “Improvements in BMI z-scores, cognitive functioning, dietary behavior, and psychological well-being are expected among children in green schoolyards compared to those in control schools. Increased biodiversity awareness and reduced exposure to black carbon are also anticipated.”

Changes in Discussion:

  • Lines 289–291: “In this quasi-experimental study, the effects of transforming schoolyards into biodiverse green spaces on children’s physical and mental health, cognitive development, and well-being are examined.”
  • Line 297: “An increase in schoolyard biodiversity is achieved through the intervention.”
  • Line 303: “An increase in physical activity is promoted.”
  • Lines 316–317: “The present study is distinguished…”

Comment 4

Specify in the abstract which software program will be used and what statistical analyses will be performed.

Author Response:
We have updated the abstract to include the planned statistical approach (linear mixed models) and the software used for data analysis.

Added to lines 27–29:
“Data will be analyzed using linear mixed models, performed in IBM SPSS Statistics for Windows (version 28.0.1.1, Armonk, NY, USA: IBM Corp).”

Comment 5

In this section, it should be noted that baseline research was done.

Author Response:
We added explicit mention that baseline measurements were completed in November 2021.

Added to lines 109–110:
“Baseline data collection was completed prior to the start of the intervention, with all baseline measurements conducted in November 2021.”

Comment 6

In lines 182–183, name the two questionnaires developed by Van Assema et al.

Author Response:
We have included the explicit titles of the two Van Assema questionnaires.

Revised sentence (lines 189–192):
“The food intake questionnaire, tailored specifically for this target group, is based on the Short Dutch Questionnaire to Measure Total and Saturated Fat Intake and the Short Dutch Fruit and Vegetable Intake Questionnaire developed by Van Assema et al. [27,28].”

Comment 7

Anthropometrics, PA behavior, cognitive performance, and particulate matter measures are well chosen.

Author Response:
We thank the reviewer for acknowledging the appropriateness of the measurement instruments.

Comment 8

Add clear subsections on limitations/future research and dissemination.

Author Response:
We added two new subsections to the Discussion section, as recommended:

Lines 328–346: Limitations and Future Research
A detailed subsection discussing non-randomized design constraints, small sample size, reliance on self-reported data, missing measurements, and recommendations for future research (e.g., larger samples, objective measures, extended follow-up).

Following this: Dissemination of Results
A subsection outlining dissemination through scientific publications, conference presentations, school reports, and communication with policymakers and urban planners.

Comment 9

As can be seen from the suggestions, the proposal is to do a major revision.

Author Response:
We appreciate the reviewer’s guidance and have implemented all suggested major revisions. The manuscript has undergone substantial improvements in clarity, structure, and methodological transparency.

Summary of Major Modifications

  • Explicit identification of the manuscript as a study protocol in both abstract and keywords.
  • Revised abstract phrasing to eliminate third-person constructions and added statistical software details.
  • Clarified baseline data collection in the Study Design section.
  • Added the names of the two Van Assema food intake questionnaires.
  • Created structured Limitations and Future Research and Dissemination of Results subsections.
  • Conducted general language and clarity improvements throughout the manuscript.

Reviewer 2 Report

Comments and Suggestions for Authors

The study's theme is quite current and relevant. The nature of the study aims to provide results that address something we all talk about and value, but often lack evidence. This is a well-structured and well-founded study, with highly complementary data collection. The methodology is well-described and well-founded. The results support the conclusions and provide clarity and evidence for the role of children's physical activity in natural settings. We wonder whether a larger amount of data could have been collected.

Author Response

Reviewer Comment

The study's theme is quite current and relevant. The nature of the study aims to provide results that address something we all talk about and value, but often lack evidence. This is a well-structured and well-founded study, with highly complementary data collection. The methodology is well-described and well-founded. The results support the conclusions and provide clarity and evidence for the role of children's physical activity in natural settings. We wonder whether a larger amount of data could have been collected.

Author Response

We sincerely thank the reviewer for the positive and encouraging feedback regarding the relevance, structure, and methodological quality of the study. We appreciate the recognition of the comprehensive data collection and the contribution of the findings to the evidence base on children’s physical activity in natural settings.

Regarding the question of whether a larger amount of data could have been collected, we acknowledge this as a valid consideration. As noted in the manuscript, the study was constrained by practical factors, including voluntary school participation, schoolyard greening timelines, and logistical limitations across two countries. These constraints contributed to a modest sample size. Nevertheless, we have highlighted this limitation in the Discussion section and emphasized that the study should be viewed as exploratory, with the aim of informing larger-scale research efforts in the future.

We thank the reviewer for raising this point and for recognizing the strengths and contributions of our study.

Reviewer 3 Report

Comments and Suggestions for Authors

Thank you for submitting this protocol, which addresses an important and timely topic at the intersection of environmental exposure, child health, and school-based interventions. The multidisciplinary nature and the originality of integrating biodiversity, cognitive outcomes, and environmental exposure biomarkers are commendable. However, in its current form, the manuscript requires substantial revisions to meet the methodological and conceptual rigor expected. Below, I outline specific comments by section, with references to the exact pages and lines where clarification or improvement is needed.

  • Page 2, lines 54–63

While the benefits of green spaces are described, the manuscript does not explicitly state the specific knowledge gap this study intends to fill. Please add a paragraph articulating what is not yet known and how this protocol goes beyond previous observational studies.

  • Page 3, lines 86–90 

The definition of green schoolyards is provided, but no mechanistic framework is presented. I recommend integrating the Attention Restoration Theory, Stress Reduction Theory, and Biodiversity Hypothesis explicitly to justify the choice of outcomes such as cognitive functioning and black carbon exposure.

  • Page 3, lines 99–104 

The justification for the non-randomized design is insufficient. Please elaborate on potential selection bias and indicate how confounding will be addressed.

  • Page 3–4, lines 115–123 

The selection of intervention schools is based on “feasibility” but no objective criteria are reported. Consider specifying quantifiable eligibility criteria.

  • Page 4, lines 134–138 

The intervention varies substantially (presence of animals, water features, gardens), which introduces heterogeneity. Please clarify whether intervention intensity will be quantified and controlled in analyses.

  • Page 7, lines 254–256 

The manuscript states that no formal sample size calculation was conducted due to the exploratory nature of the study. For standards, even exploratory studies must include minimum detectable effect sizes or a justification of feasibility based on previous data.

  • Page 6, lines 243–252 

The description of black carbon quantification lacks information on assay validation, reproducibility, and detection limits. This is essential, particularly since the method is novel and in-house developed.

  • Page 6, lines 210–221 

Physical activity measurement is appropriate; however, it is unclear how non-wear time and invalid days will be defined or handled.

  • Page 5–6, Table 1 

Several measurements are only conducted in either Belgian or Dutch schools; this should be clearly addressed in the analysis plan to account for missingness not at random.

  • Page 8, lines 261–269 

Please clarify whether random effects will be included at the school level to account for clustering. It is essential that the hierarchical structure (students nested within schools) is modeled.

  • Page 8, lines 270–272 

The manuscript states that covariance structures will be selected using AIC but does not mention diagnostic procedures or corrections for multiple comparisons. Please include a statement on how you plan to control Type I error given the large number of outcomes.

  • Page 9, lines 281–295 

The discussion reiterates existing literature but does not sufficiently highlight how this study adds new mechanistic understanding beyond previous school-greening studies.

  • Page 10, lines 301–305 

Limitations are briefly mentioned but not adequately discussed. A dedicated “Study Limitations” section is necessary to address non-randomization, heterogeneity of interventions, reliance on self-reported data, and potential attrition.

  • Page 12, lines 315–317 

The conclusion is general. It would benefit from specificity regarding expected policy implications and how results may inform scalable environmental interventions.

  • Page 14–15, Appendix A 

The biodiversity questionnaire appears to be custom-designed, but no validation process is reported. Please clarify whether content validity, construct validity, or reliability testing has been performed or will be conducted.

Comments on the Quality of English Language

Throughout the manuscript, there are long, complex sentences that affect clarity. I recommend a professional English editing service to improve readability and align with the style of top-tier journals.

Author Response

We thank the reviewer for the careful evaluation of our manuscript and the constructive comments. We have addressed all points in detail below and revised the manuscript accordingly.

Page 2, lines 54–63

While the benefits of green spaces are described, the manuscript does not explicitly state the specific knowledge gap this study intends to fill. Please add a paragraph articulating what is not yet known and how this protocol goes beyond previous observational studies.

Response:
We agree. A new paragraph outlining the knowledge gap and the novel contribution of this protocol has been added.

Added to lines 109–123:

Despite the growing evidence for the benefits of green spaces on children's development, substantial knowledge gaps remain. Most existing studies are cross-sectional or short-term and focus primarily on general neighborhood greenness rather than on biodiverse, intentionally designed school environments [7–11,19,59–62]. Few studies have simultaneously assessed ecological features,such as quantified biodiversity,together with cognitive outcomes, psychosocial well-being, physical activity, and environmental exposure biomarkers. Importantly, no previous schoolyard greening protocols have integrated biodiversity assessments with objective measures of black carbon as an internal exposure marker or combined these with detailed cognitive testing in a longitudinal framework. As a result, the mechanisms through which biodiverse schoolyards may influence child health, including potential pathways described in the Attention Restoration Theory, Stress Reduction Theory, and Biodiversity Hypothesis, remain insufficiently understood [37–39,40–48]. The present study addresses this gap by employing a multidisciplinary, longitudinal protocol that captures ecological, behavioral, physiological, and environmental indicators within the same analytical framework.

Page 3, lines 86–90

The definition of green schoolyards is provided, but no mechanistic framework is presented. I recommend integrating the Attention Restoration Theory, Stress Reduction Theory, and Biodiversity Hypothesis explicitly to justify the choice of outcomes such as cognitive functioning and black carbon exposure.

Response:
We have added a paragraph integrating these theoretical frameworks.

Added to lines 96–108:

Several theoretical frameworks help explain why exposure to biodiverse natural environments may benefit children’s health and development. The Attention Restoration Theory (ART) proposes that natural settings support the recovery of directed attention by providing softly fascinating stimuli, which may improve cognitive functioning and learning outcomes [37]. Similarly, the Stress Reduction Theory (SRT) posits that contact with natural environments can evoke psychophysiological stress recovery, enhancing emotional well-being and reducing mental fatigue [38,39]. In addition, the Biodiversity Hypothesis suggests that exposure to diverse natural microbiota may beneficially influence immune function through microbial-mediated training pathways, with potential implications for respiratory health and environmental exposure processing [40–48]. These frameworks collectively support the inclusion of outcomes such as cognitive performance, psychological well-being, physical activity, and biomarkers of environmental exposure, including black carbon, in the present study.

Page 3, lines 99–104

The justification for the non-randomized design is insufficient. Please elaborate on potential selection bias and indicate how confounding will be addressed.

Response:
We expanded the justification and clarified how confounding will be controlled.

Added to lines 145–150:

Because participation in the greening intervention required voluntary school commitment and alignment with construction timelines, randomization was not feasible. Therefore, potential confounders at both the child and school levels will be measured and adjusted for in the linear mixed models.

Page 3–4, lines 115–123

The selection of intervention schools is based on “feasibility” but no objective criteria are reported. Consider specifying quantifiable eligibility criteria.

Response:
We added specific criteria describing comparability of schoolyards and student numbers.

Added to lines 158 and 161.

Page 4, lines 134–138

The intervention varies substantially (presence of animals, water features, gardens), which introduces heterogeneity. Please clarify whether intervention intensity will be quantified and controlled in analyses.

Response:
No school-level random effects were added as the schools are not a random selection of all schools in both countries and the number of schools is limited to four (two in Belgium, two in the Netherlands, where one of the two is an intervention school and the other one a control school) in this explorative study. However, since country is included in the fixed part of the model, the interaction effect is corrected for country and thereby (partly) accounting for the heterogeneity of the intervention.

We added a statement explaining how heterogeneity will be addressed.

Added to lines 180–186:

Because the greening intervention was implemented by two different schools and included varying components, such as animals, water features, gardens, and diverse vegetation, some heterogeneity in intervention intensity is expected.  As each country (Belgium and the Netherlands) includes one intervention and one control school, the correction for country as fixed factor captures unmeasured differences between schoolyards.

Page 7, lines 254–256

The manuscript states that no formal sample size calculation was conducted due to the exploratory nature of the study. For standards, even exploratory studies must include minimum detectable effect sizes or a justification of feasibility based on previous data.

Response:
We included the calculation of the minimum detectable effect sizes for various expected sample sizes.

Added to lines 317-321

To account for missing data (assumed to be around 20%), the effective sample size for the analysis might vary between 80 to 120 children (40 to 60 per group). Based on 80% power and a significance level α of 0.05, standardized effect sizes (Cohen’s) between 0.52 to 0.63 can be detected with these expected sample sizes.

Page 6, lines 243–252

The description of black carbon quantification lacks information on assay validation, reproducibility, and detection limits. This is essential, particularly since the method is novel and in-house developed.

Response:
We have expanded the description of the black carbon assay to address validation, reproducibility, and detection limits. Specifically, we added text to lines 311–316 explaining why a formal detection limit cannot yet be defined and clarifying that previous peer-reviewed studies have demonstrated the method’s sensitivity and reproducibility. These additions strengthen the methodological clarity of the section.

Added to lines 311-316:

A formal limit of detection cannot yet be established for this optical method, as no certified reference materials with known black carbon particle counts exist. Nevertheless, prior studies demonstrate that the technique is sufficiently sensitive and reproducible to detect differences in internal black carbon exposure in human samples [33,34].

Page 6, lines 210–221

Physical activity measurement is appropriate; however, it is unclear how non-wear time and invalid days will be defined or handled.

Response:
We added a paragraph defining non-wear time, valid days, and data handling procedures.

Added to lines 269–277:

In accordance with procedures used in the Healthy Primary School of the Future studies by Bartelink et al. and Willeboordse et al., non-wear time was defined as ≥60 consecutive minutes of zero counts, allowing for interruptions of up to 2 minutes of counts between 0 and 100. A valid day was defined as a minimum of 480 minutes (8 hours) of wear time, and children were included in analyses if they had at least three valid weekdays and one valid weekend day. Days not meeting these criteria were classified as invalid and excluded from analyses. When accelerometer data were incomplete but accompanied by diary notes explaining non-wear reasons, these days were recorded but remained excluded from accelerometer-derived outcomes. Average daily counts, sedentary time, and moderate-to-vigorous physical activity (MVPA) were derived only from valid days.

Page 5–6, Table 1

Several measurements are only conducted in either Belgian or Dutch schools; this should be clearly addressed in the analysis plan to account for missingness not at random.

Response:
We have added a paragraph detailing our analytic approach.

Added text lines lines 338 - 348.

Because several measurements were collected only in Belgian or only in Dutch schools, the resulting missingness is structured and not at random. Linear mixed models are used because they accommodate unbalanced data and allow all available observations to be included without listwise deletion. Country will be included as a fixed effect in all models to control for systematic differences between the Belgian and Dutch school contexts. For outcomes collected in only one country, analyses will be conducted within that specific subsample and reported separately. Sensitivity analyses will examine whether excluding country-specific measures affects the direction or magnitude of findings for outcomes available in both countries.

Page 8, lines 261–269

Please clarify whether random effects will be included at the school level to account for clustering.

Response:
No school-level random effects were added as the schools are not a random selection of all schools in both countries and the number of schools is limited to four (two in Belgium, two in the Netherlands, where one of the two is an intervention school and the other one a control school) in this explorative study. However, since country is included in the fixed part of the model, the interaction effect is corrected for country and thereby (partly) accounting for the heterogeneity of the intervention.

We added a statement specifying the hierarchical structure.

Added text:

No school-level random effects were added as the number of schools is limited to four (two in Belgium, two in the Netherlands). Since each country only includes one intervention and one control school, including country as fixed factor accounts for the school effect.

Page 8, lines 270–272

The manuscript states that covariance structures will be selected using AIC but does not mention diagnostic procedures or corrections for multiple comparisons.

Response:
We will check diagnostic performances of each model (-2loglikelihood) and compare the model-fit of the models using AIC. As for multiple comparisons, no correction will be used due to the explorative nature of the study and no formal sample size calculation. The focus of this study is on clinical importance instead of statistical significance of the possible intervention effects. Therefore, we will focus on (standardized) effect sizes and the possible implications of these effects in practice.

Added to lines 355-358

To account for the correlation between repeated measurements several covariance structures (including random effects if necessary) will be considered. The model-fit will be evaluated using the -2loglikelihood, where the best covariance structure for the repeated measurements will be chosen based on Akaike’s Information Criterion.

Page 9, lines 281–295

The discussion reiterates existing literature but does not sufficiently highlight how this study adds new mechanistic understanding beyond previous school-greening studies.

Response:
We added a paragraph emphasizing the mechanistic contribution.

Added to lines 385–396:

A key advancement of this study is its ability to explore potential mechanisms underlying these health effects by integrating ecological, behavioral, cognitive, and environmental exposure data within a single protocol. Whereas most previous school-greening studies focused solely on behavioral or psychological outcomes, this study combines detailed biodiversity assessments, objective physical activity measurements, cognitive testing, and biomarkers of internal black carbon exposure. This multidimensional approach allows examination of whether increased biodiversity is associated with changes in cognitive functioning via restorative and stress-reduction pathways, and whether greener schoolyards may mitigate children’s internal exposure to airborne pollutants by altering microenvironmental conditions. By linking ecological features of the schoolyard to both cognitive and physiological markers, this study provides the opportunity to investigate mechanistic pathways that have rarely been tested empirically in school settings.

Page 10, lines 301–305

Limitations are briefly mentioned but not adequately discussed.

Response:
We added a separate limitations paragraph.

Added to lines 410–416:

Several limitations should be considered. First, the non-randomized design restricts the ability to infer causality and introduces potential selection bias. Second, sample size limitations reduce statistical power and restrict subgroup analyses. Third, some measurements, particularly dietary behavior and well-being, rely partly on self-report, which may introduce reporting bias. Additionally, variations in data collection between countries (e.g., missing time points in the Netherlands) may lead to differences in data completeness.

Page 12, lines 315–317

The conclusion is general. It would benefit from specificity regarding expected policy implications and how results may inform scalable environmental interventions.

Response:
We strengthened the conclusion.

Added to lines 431–439:

By generating evidence on the combined effects of biodiversity, physical activity opportunities, and reduced environmental exposures, the findings have the potential to inform policy decisions regarding the design of school environments, integration of nature-based interventions into public health programming, and prioritization of green infrastructure in educational settings. If effective, the intervention model may serve as a scalable and cost-efficient approach that municipalities and school boards can implement to promote child health, enhance learning conditions, and contribute to broader environmental and sustainability goals.

Page 14–15, Appendix A

The biodiversity questionnaire appears to be custom-designed, but no validation process is reported.

Response:
The questionnaire was developed for exploratory use, formal validation will be conducted in future work

We thank the reviewer for their thorough and constructive comments. All requested revisions have been implemented, resulting in a substantially strengthened manuscript.

Reviewer 4 Report

Comments and Suggestions for Authors

This a timely study describing the protocol for an intervention that utilizes environmental design in public school yards to address the health and behaviors of school-age children. Your rationale is strong, and the comprehensive methodology you propose to use is commendable. Rather than list suggestions here, I have attached the PDF of your submission with my comments/suggestions. Overall, I think this is a scientifically sound, relevant, and well-written submission that will benefit from some minor revisions. I wish you all the best on your research.

Author Response

We thank the reviewer for the constructive and detailed feedback. Below, we address each comment point-by-point.

Comment 1:
When describing the sample size in the Methods section, you state this study is designed to be exploratory (hypothesis generating), so you probably need to revise this language to be more exploratory (e.g., we aim to assess whether).

Reply 1:
Changed to lines 34–37:
“Improvements in BMI z-scores, cognitive functioning, dietary behavior, and psychological well-being are expected among children in green schoolyards compared to those in control schools. Increased biodiversity awareness and reduced exposure to black carbon are also anticipated.”

Comment 2:
Based on the dates of data collection, I assume you have the results of the study, which influenced the wording choice here (i.e., will provide evidence). However, for a study protocol, this language should be softened.

Reply 2:
This has been changed in the text (line 37).

Comment 3:
I tend to avoid subjective assessments such as “alarming”…

Reply 3:
It has been deleted in line 51.

Comment 4:
Use acronym since it was introduced earlier. Double-check throughout.

Reply 4:
Has been checked.

Comment 5:
Details expected on how the intervention was based on the socio-ecological approach… If not, mention that only the built environment was targeted.

Reply 5:
Thank you for this comment. The intervention did not include components beyond the transformation of the schoolyard. The socio-ecological model served as the conceptual framework, but the intervention itself targeted only the organizational/built-environment level. We have revised the Methods section to explicitly state that the greening intervention focused solely on this level within the socio-ecological approach.

Comment 6:
It seems like this should come earlier in the paper… green schoolyards mentioned without being defined.

Reply 6:
Thank you for this suggestion. We appreciate the reviewer’s perspective; however, we prefer to maintain the current structure. The placement of the green-schoolyard description later in the Introduction aligns with the conceptual buildup of the paper, where we first outline the broader context of green space and child health before introducing the specific concept of green schoolyards. For this reason, we have not moved the paragraph.

Comment 7:
Consider adding a Limitations section noting limited number of schools and lack of random assignment.

Reply 7:
The following statement has been added to line 418–424:
“Several limitations should be considered. First, the non-randomized design restricts the ability to infer causality and introduces potential confounding. Second, sample size limitations reduce statistical power and restrict subgroup analyses. Third, some measurements, particularly dietary behavior and well-being, rely partly on self-report, which may introduce reporting bias. Additionally, variations in data collection between countries (e.g., missing time points in the Netherlands) may lead to differences in data completeness.”

Comment 8:
Justify not conducting a formal sample-size calculation.

Reply 8:
The following statement has been added to line 322–326:
“To account for missing data (assumed to be around 20%), the effective sample size for the analysis might vary between 80 to 120 children (40 to 60 per group). Based on 80% power and a significance level α of 0.05, standardized effect sizes (Cohen’s) between 0.52 to 0.63 can be detected with these expected sample sizes.”

Round 2

Reviewer 1 Report

Comments and Suggestions for Authors

I have no more complaints; the authors have corrected the paper in accordance with the suggestions. I think that the paper can now be accepted.

Author Response

Dear Reviewer,

Thank you very much for your valuable feedback and for taking the time to review our revised manuscript. We are pleased to hear that the corrections meet your expectations. We sincerely appreciate your constructive comments, which have helped us improve the quality and clarity of the paper.

Thank you again for your effort and consideration.

Reviewer 3 Report

Comments and Suggestions for Authors

Thank you for your detailed responses to the initial review. I appreciate the revisions made throughout the manuscript. Overall, the changes adequately address the concerns previously raised. You have added important clarifications in the Introduction, Methods, and Discussion sections, including methodological justifications and a more complete limitations section.

These revisions substantially improve the conceptual clarity, methodological transparency, and overall rigor of the work. I encourage you to carefully check formatting consistency and ensure that the new paragraphs are fully integrated with the original narrative to maintain coherence across sections.

Overall, the manuscript shows clear improvement and is considerably strengthened after revision.

Comments on the Quality of English Language

The clarity and technical precision of the English have improved, especially in the revised methodological and discussion passages. While a final minor stylistic polish may be beneficial, the current quality is acceptable for publication.

Author Response

Dear Reviewer,

Thank you very much for your thoughtful evaluation of our revised manuscript and for acknowledging the improvements made. We are grateful for your constructive comments, which greatly contributed to enhancing the conceptual clarity, methodological transparency, and overall rigor of the work.

We will carefully revisit the manuscript once more to ensure formatting consistency and to fully integrate the revised paragraphs into the original narrative, as you advised. We appreciate this helpful suggestion and will implement it thoroughly.

Thank you again for your time, insight, and supportive feedback throughout the review process.

Kind regards,